# Human Milk Oligosaccharides in Breast Milk at Two Weeks of Age in Relation to Neurodevelopment in 2-Year-Old Children Born Extremely Preterm: An Explorative Trial

**DOI:** 10.3390/nu17050832

**Published:** 2025-02-27

**Authors:** Erik Wejryd, Erik Freiholtz Jern, Giovanna Marchini, Ulrika Åden, Eva Landberg, Thomas Abrahamsson

**Affiliations:** 1Division of Children’s and Women’s Health, Department of Biomedical and Clinical Sciences, Linköping University, 581 83 Linköping, Sweden; erik.wejryd@liu.se (E.W.); erila932@student.liu.se (E.F.J.);; 2Department of Pediatrics, Vrinnevi Hospital, 601 82 Norrköping, Sweden; 3Department of Neonatology, Astrid Lindgren Children’s Hospital, Karolinska University Hospital, 171 76 Stockholm, Sweden; 4Department of Women’s and Children’s Health, Karolinska Institute, 171 77 Stockholm, Sweden; 5Department of Clinical Chemistry and Department of Biomedical and Clinical Sciences, Linköping University, 581 83 Linköping, Sweden; eva.landberg@regionostergotland.se; 6Crown Princess Victoria Children’s Hospital, Linköping University Hospital, 581 85 Linköping, Sweden

**Keywords:** neonatal, preterm, breast milk, oligosaccharides, diversity, neurodevelopment, cognitive development

## Abstract

*Background:* Preventing neurodevelopmental impairment after extremely preterm birth remains challenging. While breast milk feeding is linked to better neurodevelopment, the underlying mechanisms are unclear. This study explored the association between individual human milk oligosaccharides (HMO) and neurodevelopment at two years of age in extremely preterm children. *Methods:* Milk samples from mothers of 76 extremely preterm infants collected at two weeks after birth were analyzed for 15 dominant HMOs. Register data from examination and Bayley-III neurodevelopmental assessment at two years’ corrected age was retrieved and categorized into levels of impairment. An exploratory analysis examined associations between the HMO composition and neurodevelopment. *Results:* Bioinformatic volcano plots revealed associations between specific HMOs and outcomes: 3FL with less neurodevelopmental impairment, LSTb with higher Bayley-III cognitive scores, and LSTa with worse neurodevelopmental impairment outcomes. Spearman correlations indicated LSTa was linked to more neurodevelopmental impairment (*p* = 0.018), lower language (*p* = 0.009), and motor (*p* = 0.02) scores, whereas 3FL correlated with less neurodevelopmental impairment (*p* = 0.02). Dichotomized analysis showed LSTa was associated with more neurodevelopmental impairment and lower language scores (*p* < 0.05), 3FL with milder neurodevelopmental impairment (*p* < 0.05), and LSTb with better cognitive (*p* < 0.01) and language (*p* < 0.05) scores. No significant associations were found for HMO diversity, total sialic acid content, or secretor/Lewis patterns. *Conclusions:* In this explorative hypothesis-generating study, certain HMOs appeared to be associated with both potentially beneficial and adverse neurodevelopmental outcomes in extremely preterm infants. However, these findings should be interpreted with caution, as they do not constitute evidence but rather serve as a preliminary foundation for future hypothesis-driven research.

## 1. Introduction

Long term neurodevelopmental impairment (NDI) among survivors after extremely preterm birth (EPT; before gestational week 28 + 0) remains a challenge despite reduced neonatal morbidity and mortality [1,2]. A Swedish study on infants born before gestational week (gw) 27 showed that 44.5% of the infants that survived to 2.5 years of age had impairments in cognition, language abilities or motor skills, compared to 14.2% among healthy full term born controls [3]. More recent data reveals that 55% of children born before 24 gw were subject of multiprofessional developmental supporting care [4]. Brain growth is restricted in EPT infants with negative consequences of later performance [5,6].

Nutrition is key for supporting brain growth and development [7]. There are evidence suggesting that breast milk feeding confers benefits for long term neurodevelopmental outcomes with reports of higher IQ, reduced incidence of white matter injury and less brain volume restriction [8,9,10,11]. Although the mechanisms are yet poorly understood, bioactive molecules in human milk have been considered important [12].

Human milk oligosaccharides (HMO) are the third most abundant nutrient in human breast milk, following lactose and lipids [13]. Over 100 distinct HMO have been identified, and their concentration vary significantly among mothers, with notable geographical variations [14]. Particular excretion patterns of secretors/non-secretors (Se+/Se−) and Lewis positive/negative subjects (Le+/Le−) determined by the genotype have been identified [13].

Nervous system development can be influenced by HMO through various mechanisms. By nourishing bifidobacteria and inhibiting pathogen adhesion, HMOs modulate the gut microbiome [15], which might reduce infection and inflammation; and impacting the gut-brain axis and brain development [16]. HMOs are rich in sialic acid, a key compound for brain maturation, serving as a direct or indirect source [17]. Notably, 3′-Sialyllactose, a sialic acid-containing HMO, is linked to language development in infancy [18]. Similarly, fucosyl lactose, another major HMO group, has been associated with cognitive and language development in full-term infants [19,20].

The aim of this exploratory study was to relate HMO levels in breast milk from mothers to extremely preterm infants 14 days after delivery, and neurological development of the child at two years of age. The hypothesis was that HMO levels and composition differ in infants with and without neurodevelopment impairment.

## 2. Materials and Methods

The present study was a part of the prospective, randomized-controlled, multi-center “Prophylactic Probiotics to Extremely Low Birth Weight Premature Infants” (PROPEL) trial evaluating the effect of probiotic *Lactobacillus reuteri* DSM 17938 on feeding tolerance, growth, severe morbidities, and mortality in ELBW premature infants (ClinicalTrials.gov ID NCT01603368, accessed on 22 February 2025). A detailed study design and the clinical outcomes have been published elsewhere [21].

### 2.1. Participants

Newborn extremely preterm (EPT) infants were enrolled between June 2012 and August 2015 at the tertiary neonatal intensive care units of Linköping University Hospital and Karolinska University Hospital in Solna, Sweden. Of the 239 eligible infants, 134 were included in the study (Figure 1). A comprehensive description of the trial protocol and neonatal outcomes has been published previously [14].

Inclusion criteria for the PROPEL study was birth in gw 23–27, birth weight ≤ 1000 g, age at inclusion less than 72 h, parents living in the catchment areas of Linköping or Karolinska university hospitals and written informed consent signed by the parents. Exclusion criteria was deadly or complicated malformation known at the time of inclusion, chromosomal abnormality known at the time of inclusion, no realistic hope of survival at the time of inclusion, gastrointestinal malformation, study product not being introduced within 72 h, or that the infant was included in other trial involving nutrition.

### 2.2. HMO Sampling and Analysis

Breast milk samples were collected from the biological mother if she had enough to exclusively feed her infant two weeks after birth. The HMO analysis has previously been thoroughly described [21]. In short, thawed breast milk underwent a series of purification steps. Initially, the milk was centrifuged to achieve removal of lipids. The infranatant was collected, and a solution containing the two oligosaccharides galacturonic acid and stachyose was introduced as an internal standard. Subsequently, proteins were precipitated out of the mixture by adding cold ethanol, followed by another round of centrifugation to remove these proteins. The supernatant underwent further purification by being passed through a C18 column designed to filter out long carbon molecules. To ensure the complete removal of any residual proteins, the eluate was centrifuged once more in an ultra-filtering tube with a molecular size cut-off at 3 kDa, maintained at 4000× *g* for 40 min. Ethanol evaporation was accelerated with pressurized air over the tube for 60 min at 40°. Before analyzing neutral oligosaccharides, a final purification step was performed by applying the sample to an anion exchange bonded silica cartridge, with water added afterwards. The eluate was then collected.

Based on previous studies, 9 neutral and 6 acidic HMOs identified as commonly being the most abundant in breast milk were selected [14] for quantification using high-performance anion-exchange chromatography (HPAEC) with pulsed amperometric detection. The HMOs with full names and abbreviations used in the text are listed in Table 1. Separate programs were used for neutral and acidic HMOs. During analysis for neutral oligosaccharides, the solution passed the HPAEC column in a constant concentration of NaOH of 20 mM and an increasing gradient of sodium acetate. In the analysis of acidic (sialylated) oligosaccharides a steady concentration of NaOH was used with the sodium acetate gradient being increased in two steps. Analyses were conducted at temperatures ranging from 20 °C to 30 °C to ensure maximal separation. Peak areas recorded during the analysis were compared to areas of internal standards with known concentrations. We also analyzed HMO oligosaccharide standards for identification and to determine response factors.

### 2.3. Neurodevelopmental Outcome Assessment

Routine follow-up after EPT birth in Sweden includes an evaluation at 24 ± 3 months corrected age [22]. This includes parents’ report on well-being, hearing and vision; physical examination, auxology, standardized neurologic examination, general assessment of gross- and fine motor function, report and grading of cerebral paresis, and whether the child is perceived to have normal general development. Psychologists performed assessment with Bayley Scales of Infant and Toddler Development, 3rd edition (Bayley-III) [23]. The collected data was then recorded in the Swedish Neonatal Quality register (SNQ), and data from this register were used when assessing neurological development in this study.

NDI was graded into four categories: none, mild, moderate, and severe. When available, index scores from the Bayley III assessment were used to assess NDI. The cut-offs adopted were consistent with those utilized in preceding Swedish research [3]. Notably, some children did not complete the full Bayley III assessment. For these individuals, evaluations were conducted collaboratively by two investigators, relying on data from relevant registries regarding diagnosed cerebral palsy, gross motor function, doctor’s report on general development, head control, sitting, walking, and fine motor function, parents report of speech and walking ability, and visual or hearing impairment in accordance with the most severe disability (Table 2).

### 2.4. Ethics

The study plan was reviewed and approved by Ethics Committee for Human Research in Linköping (Dnr 2012/28-31, 2016/503-32, 2019-04975). Written consents were obtained from the parents of all study participants.

### 2.5. Statistical Analysis

For continuous non-normally distributed data, Wilcoxon rank sum test was used when two outcomes where compared, and Kruskal-Wallis rank sum tests were used for comparing more than two groups. Adjustment for multiple comparisons with Bonferroni correction was applied for HMO background data and for HMO levels. For categorical data, Pearson’s Chi-squared test or Fisher’s exact test was applied as appropriate.

Spearman’s correlation between levels of the 15 HMOs and NDI was calculated. To further assess the data, the fold change of the HMOs was calculated, and NDI was dichotomized by merging the groups mild, moderate, and severe. To look for individual associations to the NDI outcome, a volcano plot was used, based on log2 fold change and *p* values calculated by Wilcoxon rank sum test that was adjusted for multiple comparisons. A log2 fold change above 0,6 and a *p* < 0.1 was chosen for cutoff limit for identifying associations that were to be explored further. Principal Component Analysis (PCA) was used to evaluate the beta-diversity.

Alpha diversity expressed as Shannon diversity index of the HMO composition was calculated using the vegan library in R. The sum of sialic acid containing HMOs was calculated by summarizing all HMOs with sialic acid (DSLNT counted twice since it has two sialic acids). To check if HMO levels can be used as predictor for neurodevelopmental outcomes ROC-curves for dichotomized NDI (none versus mild, moderate, and severe) were created and AUC calculated.

All statistical analyses were made in R version 4.3.1 from The R Foundation.

## 3. Results

A total of 76 extremely preterm infants were included in this study (Figure 1). The background data, such as, perinatal and family factors and neonatal complications are displayed in Table 3. A drop out analysis did not show any differences between infants included and not included in the final analyses (Appendix A).

The gestational age at birth, birth weight z-score, the number of days in respirator, NEC, and bronchopulmonary dysplasia (BPD) were associated with a worse NDI outcome at two years of age (Table 3).

Median levels of Shannon alpha diversity index for HMO composition were not significantly associated with neurodevelopmental outcomes and hence no further adjustment for multiple comparisons was performed for this analysis (Figure 2).

Neither did the beta diversity differ. The PCA plots did not separate subjects based on HMO levels (Figure 3). Mothers’ genotype for secretor and Lewis status was indirectly determined by the presence of 2FL, LDFT, LNFP I, LNFP II and LNDH I in the milk samples (Table 1). The secretor and Lewis status were clearly separated in the PCA-plot (Figure 3) but were not significantly associated with neurodevelopmental outcome. Out of the twelve children without NDI, all had mothers that were Lewis positive (Le+), while 83% (53/64) of infants with NDI had Le+ mothers (*p* = 0.20). Nine out of twelve children without NDI (75%) had secretor positive (Se+) mothers, while among the children with NDI 78% (50/64) had Se+ mothers were secretor positive (*p* = 1.0). Among infants with a severe NDI, all had mothers who did not secrete Lacto-*N*-difucohexaose I (LNDH I), reflecting that they were either non-secretors, Lewis negative or both (Table 1).

For analysis of associations between individual HMOs and the neurodevelopmental outcomes, we performed a stepwise analysis. First volcano plots were used for the dichotomized outcomes of NDI and the three Bayley-III domains to visualize the associations with median HMO levels. Bonferroni adjustment for multiple comparisons was applied and cognition was associated with higher LSTb and NDI with lower 3FL and higher LSTa (Figure 4). Those HMOs with significant associations to neurodevelopmental outcomes (3FL, LSTa and LSTb) were picked for further pair-wise analyses of the HMO concentrations and neurodevelopmental outcomes (Figure 5). In addition to the observations in the volcano plot, the Bayley III language index score was seen to be negatively correlated with LSTa and positively with LSTb.

Possible confounders related to NDI (gestational age, birth weight, necrotizing enterocolitis, days in ventilator and bronchopulmonary dysplasia) were not significantly associated with LSTa, LSTb and 3FL.

Furthermore, associations between levels of each HMO and the degree of NDI were analyzed (Table 3). The levels of the 15 different HMOs varied greatly. There was an observation that Lacto-N-difucohexaose I (LNDH I) was lower in milk to infants who developed severe NDI, but after adjustment for multiple comparisons no significant differences remained.

An exploratory analysis was conducted using Spearman correlations between the concentration of HMOs and the outcomes NDI and the three different Bayley III index scores (cognition, language, and motor). The results showed that LSTa correlated with a less favorable NDI outcome and with lower scores on both the Bayley III language and motor index scores. In contrast, 3FL correlated with a better NDI outcome. After Bonferroni adjustment for multiple comparisons no correlations remained significant (Figure 6). For all other HMOs no significant correlations were found. For the full exploratory correlation analysis, see Appendix A.

All analyzed sialylated HMOs where summarized (DSLNT was counted twice since it has two sialic acid groups) and compared against NDI and the three different Bayley scale index scores. There were no correlations between the amount of sialic acid and neurological development (Figure 7).

## 4. Discussion

In this exploratory study on extremely preterm born infants, we evaluated the association between HMO content in mother’s milk and neurodevelopment. We found that the levels of certain HMOs occasionally were associated with NDI and Bayley III index scores.

The association between 3FL and less NDI is in line with previous findings that high exposure of 2FL and 3FL was associated with improved connectivity and myelination in term born infants [24]. Another study showed that the total amount of fucosylated HMOs, of which 2FL and 3FL are the most abundant, an infant was exposed to at six months of age was positively associated with language development at 18 months [20]. While 2FL exposure at one month has been linked to cognitive development at 24 months [24], our study did not observe improved neurodevelopmental outcomes associated with 2FL. The precise mechanisms by which fucosylated HMOs influence neurodevelopment remain unclear. They may act indirectly by modulating the gut microbiome and gut-brain axis or directly through bacterial metabolites that support cognitive development [19]. The findings in our, and the mentioned studies, suggest that fucosylated HMOs are associated with neurodevelopment.

Among the 15 analyzed HMOs, LSTa was least abundant in the milk samples, but at the same time LSTa was associated to a worse NDI outcome. No previous research that associates LSTa with cognitive development has been reported. On the other hand, previous research has identified a correlation between LSTb levels and impaired cognitive development [21]. In contrast to the previous research, we found that LSTb was related with improved cognitive function and language skills at two years of age. The conflicting results indicate that the functions of HMOs are complex. Further studies are needed to elucidate the role of the LST-family of HMOs.

Earlier hypotheses suggesting the need for sialic acid supplementation in infant feeding, supported by studies showing cognitive impairment in sialic acid-deprived mice, were not confirmed in our trial [17,25]. We found no association between sialic acid levels in breast milk and neurodevelopmental outcomes. This does not necessarily invalidate previous research on the link between sialic acid and neurodevelopment, as the breast milk provided in our study likely contained sufficient sialic acid to meet the basic requirements for normal neurodevelopment.

Given the considerable variation in HMOs between different mothers and across geographical regions, we considered it worth exploring whether the composition or diversity of HMOs plays a meaningful role in infant health and development, as has been suggested previously [14]. For instance, previous research has linked greater HMO diversity to a lower incidence of NEC in extremely preterm infants [21]. There are no previous studies that have examined the association between HMO diversity and neurodevelopment. In our study we did not find that the diversity was associated with NDI or the Bayley III index scores. Previous research has suggested that the substantial variation in HMO composition may represent an adaptive maternal mechanism to support optimal conditions for infant growth and development [26]. Consequently, attributing developmental impairments to specific HMO levels or compositional patterns may be of limited validity. The development of NDI is also multi-factorial and EPT infants are at risk of serious neonatal complications which might overpower the effects of HMO.

The concentrations and types of HMOs present in a mother’s breast milk are predominantly influenced by her secretor and Lewis genotype [27]. When analyzing the data using PCA plots, the only pattern we found was that the secretion type (secretor or Lewis) affected the diversity of the samples. Interestingly, we found that all infants that did not develop NDI had mothers who were Lewis positive, although this finding was not statistically significant.

### Strengths and Limitations

Our study had several strengths. One of them is that the infants were fully fed with the mother’s milk at the time of milk sampling, and did not receive donor milk or formula. Another strength is the fact that the PROPEL study had two inclusion sites which increases the possibility that the study sample is representative for the patient group. Furthermore, only extremely preterm born infants were included, which makes it easier to make conclusions about this patient group that have a high risk of acquiring neurodevelopmental impairment, although the results in this trial should not be considered generalizable for all preterm infants.

A total of 32 individuals was lost since HMO data when the infant was two weeks old were absent in the PROPEL study but drop out analysis did not indicate any bias being induced by this.

There are several limitations in this trial. Firstly, numerous medical and social factors can influence neurodevelopment in preterm infants during their early years. Due to the limitations of our inclusion data and register-based follow-up, we were only able to explore a limited number of these potential confounding factors.

Secondly, we quantified HMO concentrations in collected samples rather than assessing the infant’s total cumulative HMO intake. Additionally, we measured HMO concentrations at a single time point. Given that HMO secretion patterns fluctuate over time, the quantifications in this study may not fully capture the infant’s overall HMO exposure. However, previous research by our group has identified consistent patterns in individual excretion over time which helps mitigate the impact of this limitation [21].

Another weakness is that not all children did undergo a full Bayley III assessment at the age of two years. For those children the neurological outcome instead has been evaluated based on register data. This method has previously been used but is not a validated method and we did not perform a sensitivity analysis [3]. This is to a lesser extent based on objective findings and systematic examination and could possibly cause this subgroup to be reported having a less impairment since subtle difficulties may pass unnoticed. This reduces the possibility for this study to accurately identify associations between HMO exposure and neurodevelopmental outcomes.

The study was exploratory, and we did not have a defined hypothesis for each individual analysis performed. The high number of analyses performed imputes a risk of statistically significant findings occurring by chance. The adjustment for multiple comparisons using the conservative Bonferroni correction before choosing which HMOs to further analyze after the initial volcano plot should address this. The sample size of 76 analyzed infants may be considered relatively small, particularly given the large number of analyses performed and the corrections for multiple comparisons, both of which reduce the likelihood of detecting true associations.

Considering this, our findings should entirely be viewed as hypothesis-generating and only further studies on the topic can deliver evidence of correlations. This present study does not give any answers regarding mechanisms or causality.

## 5. Conclusions

In this exploratory study, we observed potential associations between the levels of certain HMOs in breast milk and neurodevelopmental outcomes in children born extremely preterm. Specifically, 3FL and LSTb appeared to be linked to more favorable neurodevelopmental outcomes, whereas LSTa was associated with a higher occurrence of NDI and poorer language outcomes. However, no associations were found between neurodevelopment and maternal HMO secretor or Lewis status, total sialic acid content, or overall HMO diversity. Given the broad scope of this study and the absence of a predefined hypothesis, these findings should only be interpreted as a foundation for future hypothesis-driven research.

## Figures and Tables

**Figure 1 nutrients-17-00832-f001:**
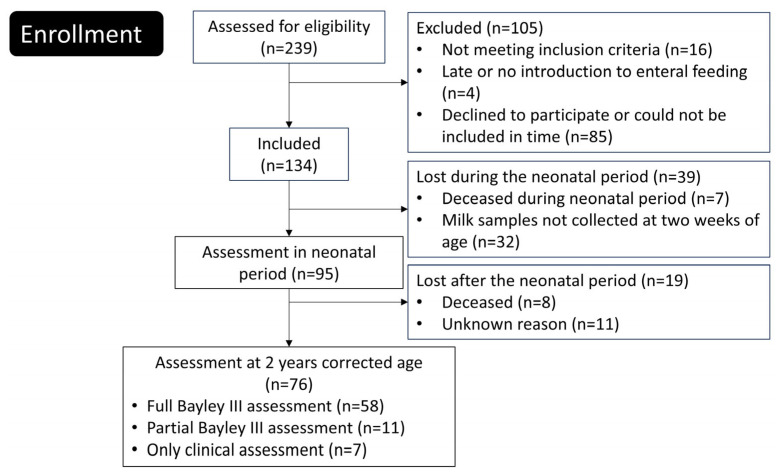
Flowchart of the study.

**Figure 2 nutrients-17-00832-f002:**
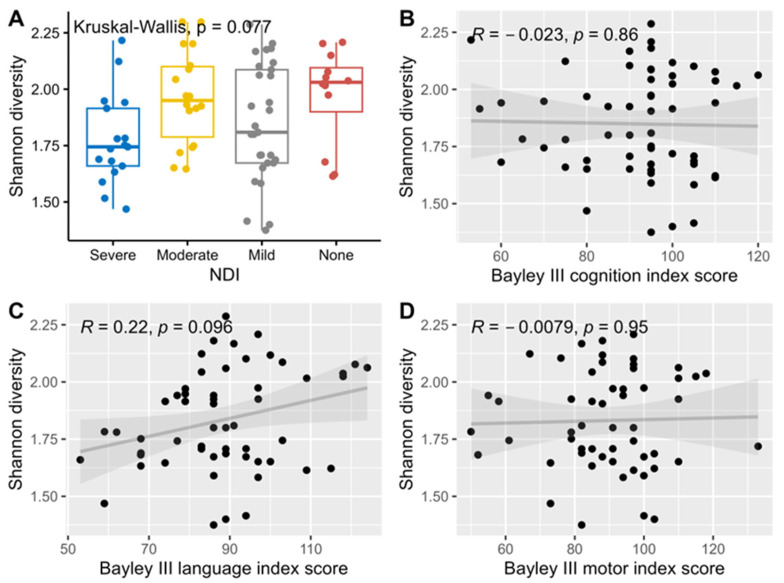
Association and correlation between Shannon diversity index, NDI (**A**), and Bayley III index scores, cognition (**B**), language (**C**) and motor (**D**) using Kruskal-Wallis rank sum tests for NDI, and Spearman correlations for the three Bayley III index scores.

**Figure 3 nutrients-17-00832-f003:**
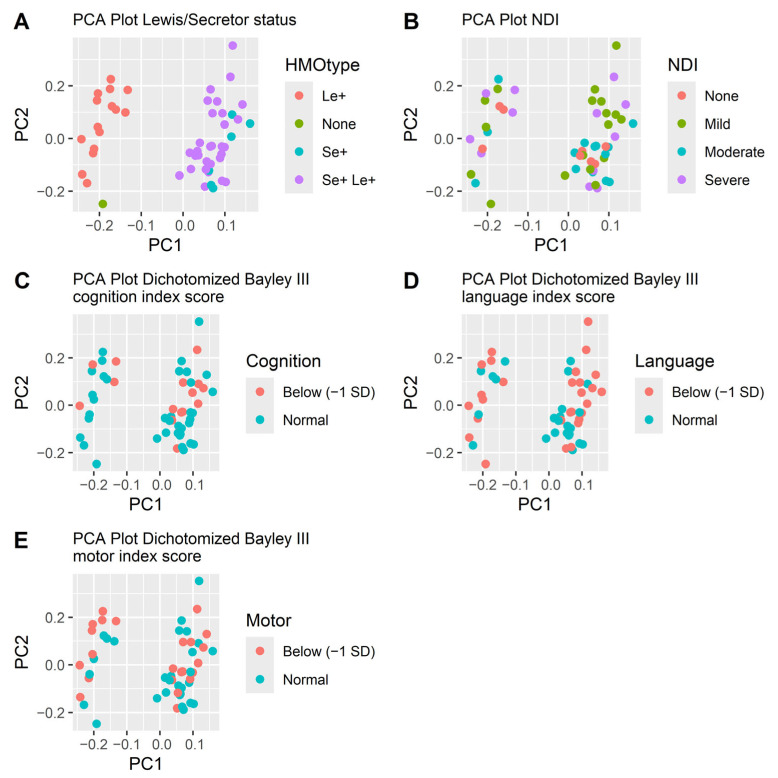
PCA plots of the HMOs at week two colored by how the HMOs are secreted (**A**), NDI (**B**) and the three dichotomized Bayley III index scores cognition (**C**), language (**D**) and motor (**E**). The separation follows the secretion pattern dictated by the mother’s secretor or Lewis genotype (**A**) but is not associated with neurodevelopmental outcomes (**B**–**E**).

**Figure 4 nutrients-17-00832-f004:**
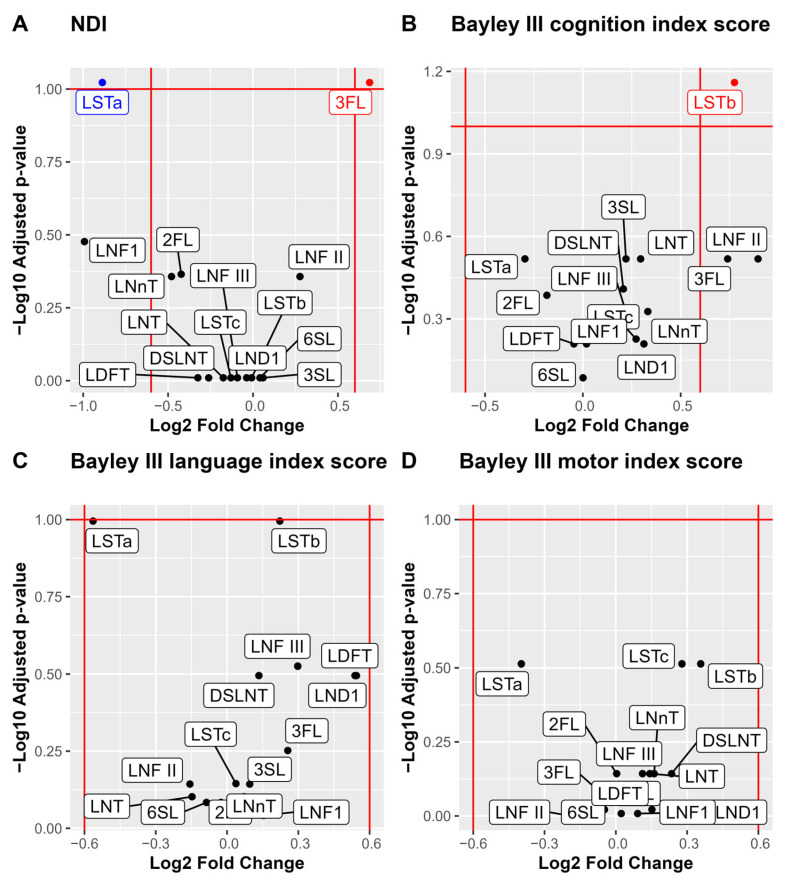
Volcano plots and boxplots of NDI and the three Bayley III index scores cognition, language, and motor. NDI was dichotomized by none versus, mild, moderate, and severe, and the three Bayley III index score were dichotomized by –1 SD. Volcano plots show NDI (**A**), Bayley III index scores cognition (**B**), language (**C**) and motor (**D**). *p* values are calculated using Wilcoxon rank sum test and Bonferroni adjusted for multiple comparisons. Significance level *p* < 0.1. Log2 fold cut offs ± 0.6.

**Figure 5 nutrients-17-00832-f005:**
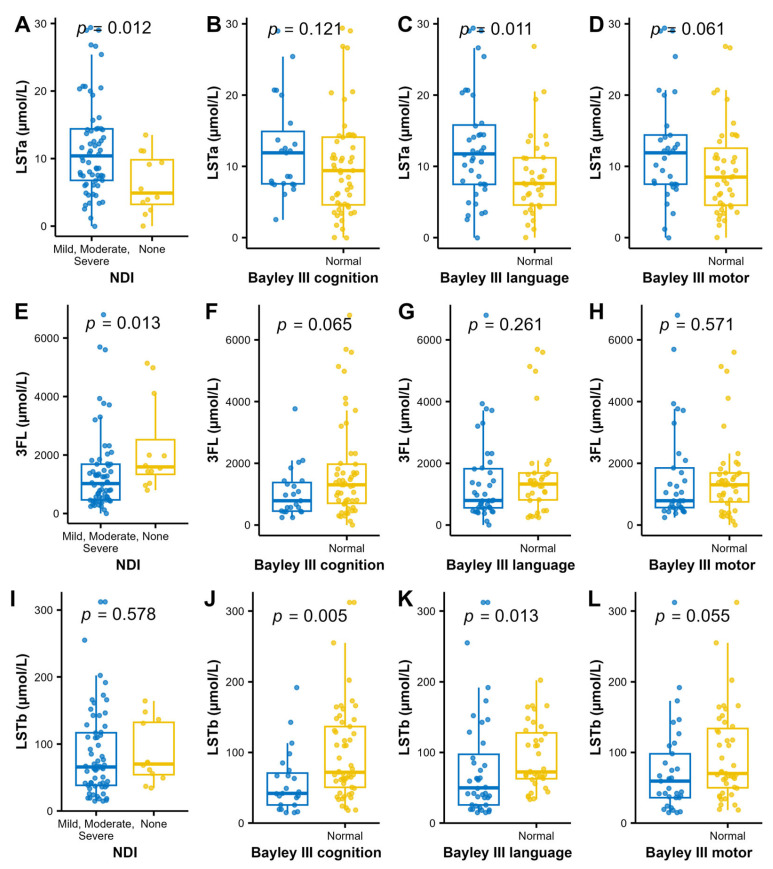
Levels of LSTa (**A**–**D**) 3FL (**E**–**H**) and LSTb (**I**–**L**); and the dichotomized NDI and Bayley III index to further analyze the findings in the bioinformatic volcano plots (Figure 4). Wilcoxon rank sum test was used for analyzing the difference in distribution of HMO concentrations between the normal and impaired groups.

**Figure 6 nutrients-17-00832-f006:**
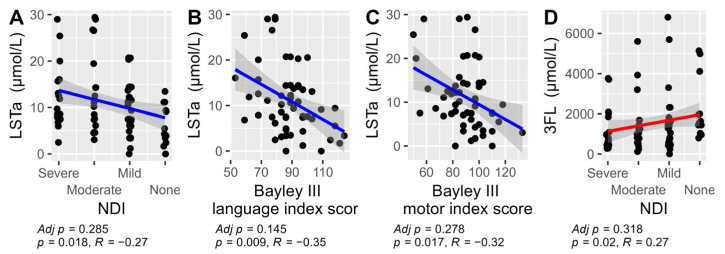
The significant Spearman correlations between HMOs and their total sum versus NDI and the three Bayley III index scores: cognition, language, and motor. Results were not adjusted for multiple comparison in this exploratory analysis. LSTa versus NDI (**A**), LSTa vs Bayley III language index score (**B**), LSTa versus Bayley III motor index score (**C**) and 3FL versus NDI (**D**). The non-significant correlations are not displayed (see Appendix A).

**Figure 7 nutrients-17-00832-f007:**
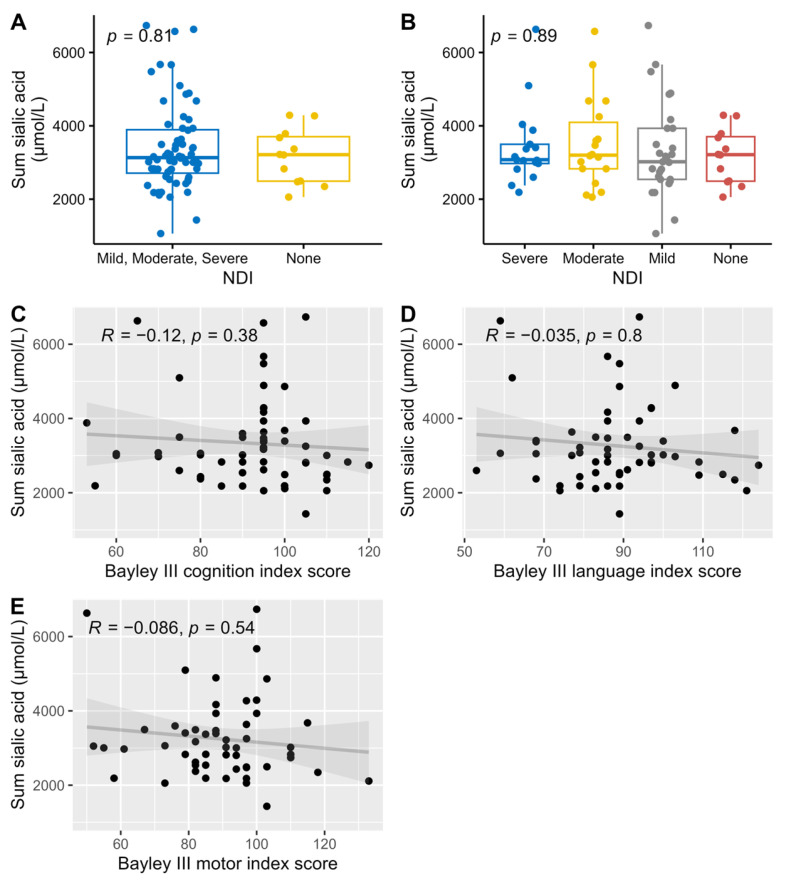
Association and correlations between the summary of all sialylated HMOs versus dichotomized NDI (**A**), NDI (**B**), and Bayley III index scores, cognition (**C**), language (**D**), and motor (**E**), with associations Wilcoxon (**A**) and Kruskal-Wallis (**B**) tests used for NDI, and Spearman correlations for the Bayley III index scores.

**Table 1 nutrients-17-00832-t001:** Median human milk oligosaccharides (HMO) levels (µmol/L) in breast milk from mothers to extremely preterm born infants collected at week two.

Name	Type	Secreted by	Neurodevelopmental Impairment					
			None * N * = 12 ^1^	Mild * N * = 29 ^1^	Moderate * N * = 18 ^1^	Severe * N * = 17 ^1^	* p * -Value ^2^	q-Value ^3^
3′-sialyllactose (3SL)	Sialylated	All	306 (249–386)	286 (256–352)	383 (296–444)	331 (236–357)	0.20	>0.90
6′-sialyllactose (6SL)	Sialylated	All	1203 (1000–1490)	1174 (869–1442)	947 (814–1430)	1368 (1076–1653)	0.30	>0.90
Sialyl-lacto-N-tetraose a (LSTa)	Sialylated	All	5 (3–10)	11 (6–14)	11 (6–18)	10 (8–16)	0.065	>0.90
Sialyl-lacto-N-tetraose b (LSTb)	Sialylated	All	70 (54–133)	67 (36–109)	83 (60–117)	44 (33–113)	0.40	>0.90
Sialyl-lacto-N-neotetraose c (LSTc)	Sialylated	All	127 (86–200)	127 (88–150)	148 (59–200)	146 (73–182)	>0.90	>0.90
Disialyl-lacto-N-tetraose (DSLNT)	Sialylated	All	594 (564–746)	673 (457–1070)	715 (542–1017)	572 (414–942)	0.60	>0.90
2′-fucosyllactose (2FL)	Neutral	Se+	4689 (2608–6290)	5844 (4436–7661)	6105 (4027–7654)	6266 (0–8026)	0.50	>0.90
3′-fucosyllactose (3FL)	Neutral	All	1594 (1339–2524)	1282 (571–1638)	1153 (620–1622)	669 (427–1850)	0.063	>0.90
Lacto-difucotetraose (LDFT)	Neutral	Se+	445 (287–604)	442 (298–684)	504 (299–928)	279 (0–697)	0.50	>0.90
Lacto-N-tetraose (LNT)	Neutral	All	2230 (1780–2365)	1752 (1252–2704)	2489 (2129–2964)	2388 (1425–3508)	0.20	>0.90
Lacto-N-neotetraose (LNnT)	Neutral	All	133 (50–238)	205 (134–257)	238 (148–396)	143 (111–214)	0.082	>0.90
Lacto-N-fucopentaose I (LNFP I)	Neutral	Se+	613 (315–1156)	1203 (640–1598)	1727 (910–2213)	1208 (0–2572)	0.13	>0.90
Lacto-N-fucopentaose II (LNFP II)	Neutral	Le+	644 (359–767)	367 (124–693)	423 (280–967)	266 (0–620)	0.30	>0.90
Lacto-N-fucopentaose III (LNFP III)	Neutral	All	349 (310–438)	363 (296–551)	448 (250–550)	346 (218–436)	0.70	>0.90
Lacto-N-difucohexaose I (LNDH I)	Neutral	Se+ Le+	942 (318–1094)	840 (0–1166)	1269 (252–1555)	0 (0–972)	0.033	0.50
Σ analyzed HMO			15,527 (12,043–16,619)	15,321 (13,008–16,869)	16,913 (14,705–19,860)	15,746 (12,998–16,829)	0.30	>0.90

All analyzed HMOs with full names, abbreviations, classification into sialylated or neutral, and classification according to the genotype the mother needs to have to produce the HMO. Se+ = secretor, Le+ = Lewis positive. ^1^ Median (inter quartile range), ^2^ Kruskal-Wallis rank sum test for analyzing association between HMO concentration and grade of neurodevelopmental impairment, ^3^ Bonferroni correction for multiple testing.

**Table 2 nutrients-17-00832-t002:** Grading of neurodevelopmental impairment (NDI) using Bayley-III and/or clinical data.

Developmental Domain	Neurological Developmental Impairment			
	None (0)	Mild (1)	Moderate (2)	Severe (3)
Cognition or language				
Bayley III cognitive scales	Bayley III Scores > −1 SD (Index ≥ 95)	Bayley III Scores −1 SD to −2 SD (Index 83–94)	Bayley III Scores −2 SD to −3 SD (Index 72–82)	Bayley III Scores < −3 SD (Index < 72)
Bayley III language scales	Bayley III Scores > −1 SD (Index ≥ 97)	Bayley III Scores −1 SD to −2 SD (Index 85–96)	Bayley III Scores −2 SD to −3 SD (Index 72–84)	Bayley III Scores < −3 SD (Index < 72)
Parent report	Speech: Uses sentences with 2–3 words	Speech: Says a few words (vocabulary > 10 words)	Speech: Says a few words (vocabulary < 10 words)	Speech: Does not speak at all
or Motor function				
Cerebral palsy (Gross motor function classification scale)	None	1	2–3	4–5 (non-ambulant)
Bayley III fine and gross motor scales	> −1 SD (Index ≥ 94)	Scores −1 SD to −2 SD (Index 80–93)	Scores −2 SD to −3 SD (Index 66–79)	Scores < −3 SD (Index < 66)
Doctor’s report	Normal head control Sits without support Walks without support Normal finger movements	Fine motor function: Clumsy but can grasp or Abnormal neurology or motor development	Fine motor function: Clumsy but can grasp	Unsteady head control Unsteady sitting or cannot sit. Fine motor: Does not grasp objects.
or Hearing				
Parent report	No hearing impairment	No hearing impairment	Hearing impairment or Has hearing aids but still impaired hearing	Cannot hear
or Vision				
Parent report	No visual impairment	Visual impairment: No impairment when wearing glasses	Visual impairment: Serious visual impairment that remains in spite of glasses or Admitted to center for visual impairments	Visual impairment: Bilateral blindness

**Table 3 nutrients-17-00832-t003:** Background characteristics for infants participating in the follow up at two years corrected age.

	Neurological Developmental Impairment				
Characteristic	None, *N* = 12 ^1^	Mild, *N* = 29 ^1^	Moderate, *N* = 18 ^1^	Severe, *N* = 17 ^1^	*p*-Value
**Perinatal data**					
Gestational age, weeks	25.9 (25.2–25.9)	26.3 (25.1–26.6)	24.6 (23.9–25.6)	25.1 (24.3–25.9)	**0.028** ^2^
Birth weight, g	789 (677–932)	803 (686–906)	676 (601–807)	696 (631–817)	**0.033** ^2^
Birth weight z-score	−0.5 (−1.6–0.0)	−0.7 (−1.1–−0.4)	−0.7 (−1.3–−0.4)	−0.6 (−1.5–−0.2)	>0.90 ^2^
Birth length z-score	−0.9 (−1.6–−0.3)	−0.7 (−1.8–0.4)	−1.3 (−2.5–−0.5)	−0.7 (−2.0–−0.4)	0.40 ^2^
Birth head circumference z-score	−0.4 (−0.9–0.1)	−0.8 (−1.3–−0.4)	−0.6 (−1.0–−0.3)	−0.7 (−1.3–−0.2)	0.50 ^2^
Apgar at 5 min ^5^	7.5 (5.8–9.0)	7.0 (4.0–8.0)	6.0 (4.0–7.0)	5.0 (3.8–7.3)	0.30 ^2^
Apgar at 10 min ^5^	8.5 (7.8–9.3)	8.0 (7.0–9.0)	8.0 (7.0–8.8)	8.0 (7.0–9.0)	0.50 ^2^
Small for gestational age	2 (17%)	7 (24%)	2 (11%)	3 (18%)	0.70 ^3^
Female sex	6 (50%)	13 (45%)	7 (39%)	6 (35%)	0.90 ^4^
Infants from multiple pregnancy	3 (25%)	12 (41%)	6 (33%)	5 (29%)	0.80 ^3^
Chorioamnionitis	2 (17%)	7 (24%)	3 (17%)	3 (18%)	>0.90 ^3^
Caesarean section	6 (50%)	18 (62%)	11 (61%)	10 (59%)	>0.90 ^3^
Maternal smoking at inclusion	1 (8.3%)	0 (0%)	1 (5.6%)	0 (0%)	0.30 ^3^
Prenatal steroids administered	11 (92%)	28 (97%)	18 (100%)	17 (100%)	0.50 ^3^
Inclusion site					0.093 ^3^
Linköping	2 (17%)	7 (24%)	10 (56%)	6 (35%)	
Stockholm	10 (83%)	22 (76%)	8 (44%)	11 (65%)	
**Neonatal complications**					
Sepsis, culture positive	3 (25%)	9 (31%)	4 (22%)	9 (53%)	0.30 ^3^
Intracerebral hemorrhage, grade 3–4	0 (0%)	3 (10%)	3 (17%)	5 (29%)	0.20 ^3^
Periventricular leukomalacia	0 (0%)	1 (3.4%)	1 (5.6%)	2 (12%)	0.60^3^
Necrotizing enterocolitis, grade II-III	0 (0%)	2 (6.9%)	0 (0%)	5 (29%)	**0.014** ^3^
Days in ventilator, total	7 (2–14)	6 (0–18)	31 (10–34)	21 (18–32)	**<0.001** ^2^
Bronchopulmonary dysplasia	4 (33%)	14 (48%)	13 (72%)	13 (76%)	**0.046** ^4^
**Family**					
Postgraduate parents, number of					0.14 ^3^
0	2 (17%)	4 (14%)	4 (22%)	7 (41%)	
1	3 (25%)	1 (3.4%)	0 (0%)	2 (12%)	
2	1 (8.3%)	10 (34%)	3 (17%)	4 (24%)	
Missing data	6 (50%)	14 (48%)	11 (61%)	4 (24%)	

^1^ Median (25–75%); n (%), ^2^ Kruskal-Wallis rank sum test, ^3^ Fisher’s exact test, ^4^ Pearson’s Chi-squared test, ^5^ One datapoint is missing for the Apgar score at both 5 and 10 min in the NDI severe group. Values in the *p*-value column that are less than 0.05 are shown in bold

## Data Availability

Dataset available on request from the authors.

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
