# Peer review of "Human Milk Oligosaccharides in Breast Milk at Two Weeks of Age in Relation to Neurodevelopment in 2-Year-Old Children Born Extremely Preterm: An Explorative Trial"

_nutrients, 2025, doi:10.3390/nu17050832_

Round 1

Reviewer 1 Report

Comments and Suggestions for Authors

Areas for Improvement:

  • Statistical Analysis:
    • The authors mention using Bonferroni correction for multiple comparisons in some instances (HMO background data, volcano plots) but state, "No other correction for multiple comparison were made." This is a significant concern. Given the number of HMOs analyzed and the multiple outcome measures, robust correction for multiple comparisons (e.g., False Discovery Rate (FDR)) is essential to control the family-wise error rate and minimize the risk of false positive findings. The lack of consistent correction weakens the statistical strength of the findings. This needs to be addressed throughout the results section.
    • The justification for not adjusting for multiple comparisons in certain exploratory analyses is unclear and requires further explanation. Exploratory analyses still benefit from some form of multiple comparison control, even if it's less stringent than confirmatory analyses. The authors should clarify their rationale.
    • The choice of a p-value < 0.1 for the volcano plot, while stated, is not a standard threshold and requires justification. Typically, a p-value < 0.05 is used. The authors should explain their reasoning behind this choice.
    • The description of the statistical methods is sometimes vague. For example, "Wilcoxon rank sum test was used for all tests in E-P." Which specific comparisons were made? More detail is needed. The same applies to the Kruskal-Wallis test. The reader needs to be able to reproduce the analysis.
    • The rationale for choosing specific HMOs (3FL, LSTa, LSTb) for further analysis (boxplots) after the volcano plot analysis is not clearly explained. Was this pre-planned, or was it based on the volcano plot results? This should be clarified to avoid any impression of data dredging.
    • The ROC curve analysis and AUC interpretation are quite limited. While the AUC of 0.80 is mentioned, the statistical significance of the predictive models is not discussed adequately. Were the models statistically significant? What were the sensitivities and specificities at relevant cut-off points? More in-depth analysis and interpretation of the ROC curves are warranted.
  • HMO Analysis:
    • While the HMO analysis method is referenced, a brief summary of the specific HMOs quantified and their abbreviations used in the figures and tables would be helpful for the reader. A table summarizing the full names and abbreviations of each HMO would significantly improve readability.
    • The rationale for selecting the 15 HMOs for analysis should be stated more explicitly. Why these 15, and not others?
  • Neurodevelopmental Assessment:
    • The method for categorizing NDI for children who did not complete the full Bayley-III assessment is a potential source of bias. While the authors acknowledge this limitation, they should discuss the potential impact of this approach on the results in more detail. How might this have influenced the observed associations? Was any sensitivity analysis performed to assess the impact of this imputation method?
    • The reliance on parent reports and doctor's reports for some children's NDI assessment introduces potential subjectivity. The authors should acknowledge this limitation and discuss its potential impact.
  • Presentation of Results:
    • Figure 3 is a bit crowded. Consider breaking it down into multiple figures for better clarity. The same applies to other figures.
    • The boxplot labels in Figure 3 (E-P) are difficult to read. Improve the resolution or consider a different way to present this data.
    • The description of the PCA plot (Figure 5) is too brief. What does the separation based on "Lewis and/or secretor phenotype" mean in terms of HMO composition? More detailed interpretation is needed.
    • Table 3 would benefit from a clearer explanation in the caption about what the "Secreted by" column represents.
    • The "Results" section sometimes reads more like a discussion. Focus on presenting the findings clearly and concisely in this section. Interpretation and contextualization belong in the "Discussion" section.

Limits:

  • Exploratory Nature: The authors acknowledge the exploratory nature of the study and the lack of pre-defined hypotheses for each analysis. This is an important limitation, and the findings should be interpreted cautiously. The authors should emphasize the hypothesis-generating nature of the study more strongly in the abstract and conclusion.
  • Sample Size: While 76 infants is a reasonable sample size, it's still relatively small, especially when considering the multiple subgroups and the number of HMOs analyzed. This limits the power to detect statistically significant associations, particularly after correction for multiple comparisons.
  • Confounding Factors: While the authors explored some potential confounders, it's possible that other unmeasured or unmeasured confounders could have influenced the observed associations. The observational nature of the study limits the ability to establish causality.
  • Single Time Point Measurement: The study only measured HMO levels at one time point (two weeks postpartum). HMO composition changes over time, so this single measurement may not fully reflect the infant's exposure to HMOs throughout the breastfeeding period.
  • Generalizability: The study population consists of extremely preterm infants born in Sweden. The findings may not be generalizable to other populations or infants born at different gestational ages.

Author Response

Reviewer 1

Areas for Improvement:

  • Statistical Analysis:
    • The authors mention using Bonferroni correction for multiple comparisons in some instances (HMO background data, volcano plots) but state, "No other correction for multiple comparison were made." This is a significant concern. Given the number of HMOs analyzed and the multiple outcome measures, robust correction for multiple comparisons (e.g., False Discovery Rate (FDR)) is essential to control the family-wise error rate and minimize the risk of false positive findings. The lack of consistent correction weakens the statistical strength of the findings. This needs to be addressed throughout the results section.

Thank you for bringing up this important topic, which of course has been subject of discussions in the author group. The basis for our decisions has overall been an ambition to cover a rather broad field of HMOs and neurodevelopmental outcomes. With our study design and the restricted knowledge in the field, we have not been aiming for establishing evidence for effects by HMOs, but instead to include findings which might be of interest for future research and provide a background for hypothesis-based studies. We have, however, used a robust method (Bonferroni) when we corrected for multiple correlation.  

      • Diversity fig 2: No significant differences were observed. We could add that correcting for multiple comparisons weakened the non-significant associations even more but choose not to write that out since it we considered it obvious.
      • Association between HMO levels and dichotomized neurodevelopmental outcomes (fig 3): To filter out possible correlations these large calculations were made and corrected for multiple comparisons with Bonferroni. This is written in the legend of fig 3 but has been further clarified in the text, which we will correct.
      • In the following analyses for HMOs selected after the volcano plot comparing median HMO levels with neurodevelopmental outcomes, we did not repeat the correction for multiple comparisons, as we perceived that this was already used in the first stage and that we thus have filtered out results that were most likely false positive. The reasoning has now been stated more clearly in the results section.
      • Correlation between HMO levels and neurodevelopmental outcomes (fig4): A correction for multiple comparisons has been added, as we agree with your point of view that it was inconsequent not to use it.

The justification for not adjusting for multiple comparisons in certain exploratory analyses is unclear and requires further explanation. Exploratory analyses still benefit from some form of multiple comparison control, even if it's less stringent than confirmatory analyses. The authors should clarify their rationale.

      • You are right about this, and we have had the intention to use correction for multiple comparisons and believe that the manuscript has improved after your comments.
    • The choice of a p-value < 0.1 for the volcano plot, while stated, is not a standard threshold and requires justification. Typically, a p-value < 0.05 is used. The authors should explain their reasoning behind this choice.
      • The use the cutoff p ≤ 0.1 is often used in exploratory analyses. A strict cutoff like p ≤ 0.05 may filter out potentially meaningful trends, while p ≤ 0.1 allows for detecting more candidates for further validation. Many small biologically relevant effects may not reach p≤0.05 due to variability in data, sample size limitations, or biological complexity. We thus chose the p ≤ 0.1 cutoff to ensure that interesting findings are not prematurely discarded and then proceeded with further analyses of the detected candidates.

    • The description of the statistical methods is sometimes vague. For example, "Wilcoxon rank sum test was used for all tests in E-P." Which specific comparisons were made? More detail is needed. The same applies to the Kruskal-Wallis test. The reader needs to be able to reproduce the analysis.
      • This has been clarified
    • The rationale for choosing specific HMOs (3FL, LSTa, LSTb) for further analysis (boxplots) after the volcano plot analysis is not clearly explained. Was this pre-planned, or was it based on the volcano plot results? This should be clarified to avoid any impression of data dredging.
      • This was based on the volcano plot which has been clarified
    • The ROC curve analysis and AUC interpretation are quite limited. While the AUC of 0.80 is mentioned, the statistical significance of the predictive models is not discussed adequately. Were the models statistically significant? What were the sensitivities and specificities at relevant cut-off points? More in-depth analysis and interpretation of the ROC curves are warranted.
      • We have discussed this after your comments and conclude that the role of the ROC-curves can be questioned in this kind of explorative study where prediction is not targeted. We have therefore omitted these curves from the manuscript.
  • HMO Analysis:
    • While the HMO analysis method is referenced, a brief summary of the specific HMOs quantified, and their abbreviations used in the figures and tables would be helpful for the reader. A table summarizing the full names and abbreviations of each HMO would significantly improve readability.
      • This actually exists in table 3 and I have added a reference in the text as well as a slightly elaborated table legend.
    • The rationale for selecting the 15 HMOs for analysis should be stated more explicitly. Why these 15, and not others?
      • These 15 have been found to be dominating in previous HMO research performed by co-author Landberg, and thus most likely represent HMOs with the most potential clinical impact. This has been clarified in the text.
  • Neurodevelopmental Assessment:
    • The method for categorizing NDI for children who did not complete the full Bayley-III assessment is a potential source of bias. While the authors acknowledge this limitation, they should discuss the potential impact of this approach on the results in more detail. How might this have influenced the observed associations? Was any sensitivity analysis performed to assess the impact of this imputation method?
      • Good remark. This is indeed a limitation and since we have not performed a sensitivity analysis, there is a possibility that this imputation limits the accuracy in the NDI-grading that reduces the possibility to identify true associations in our trial. This has been described more carefully in the limitations part of the discussion.
    • The reliance on parent reports and doctor's reports for some children's NDI assessment introduces potential subjectivity. The authors should acknowledge this limitation and discuss its potential impact.
      • Yes, reflections on this limitation have been added to the discussion paragraph.
  • Presentation of Results:
    • Figure 3 is a bit crowded. Consider breaking it down into multiple figures for better clarity. The same applies to other figures.
      • Thansk for the remakrks. We have now split figure 3.
    • The boxplot labels in Figure 3 (E-P) are difficult to read. Improve the resolution or consider a different way to present this data.
      • We have followed the guidelines of Nutrients. We have no such problems when reading the figure on and enlarge the manuscript returned from Nutrients. We will upload a new version for the revised manuscript and hope the resolution will be fine.
    • The description of the PCA plot (Figure 5) is too brief. What does the separation based on "Lewis and/or secretor phenotype" mean in terms of HMO composition? More detailed interpretation is needed.
      • Pending
    • Table 3 would benefit from a clearer explanation in the caption about what the "Secreted by" column represents.
      • Thanks for the remarks. We have added a notion.
    • The "Results" section sometimes reads more like a discussion. Focus on presenting the findings clearly and concisely in this section. Interpretation and contextualization belong in the "Discussion" section.
      • Thanks for the remarks. We have made adjustments, but we have kept som explanatory notion to make it easier for the reader.

Limits:

  • Exploratory Nature: The authors acknowledge the exploratory nature of the study and the lack of pre-defined hypotheses for each analysis. This is an important limitation, and the findings should be interpreted cautiously. The authors should emphasize the hypothesis-generating nature of the study more strongly in the abstract and conclusion.
    • We agree that this could be stated more clearly and have rewritten the requested parts.
  • Sample Size: While 76 infants is a reasonable sample size, it's still relatively small, especially when considering the multiple subgroups and the number of HMOs analyzed. This limits the power to detect statistically significant associations, particularly after correction for multiple comparisons.
    • Correct. Included among the limitations.
  • Confounding Factors: While the authors explored some potential confounders, it's possible that other unmeasured or unmeasured confounders could have influenced the observed associations. The observational nature of the study limits the ability to establish causality.
    • Correct. Included among the limitations.
  • Single Time Point Measurement: The study only measured HMO levels at one time point (two weeks postpartum). HMO composition changes over time, so this single measurement may not fully reflect the infant's exposure to HMOs throughout the breastfeeding period.
    • Correct. Included among the limitations.
  • Generalizability: The study population consists of extremely preterm infants born in Sweden. The findings may not be generalizable to other populations or infants born at different gestational ages.
    • This has been clarified better now.

Reviewer 2 Report

Comments and Suggestions for Authors

Comments and Suggestions on the Manuscript:

I appreciate the opportunity to review this article. I find it to be an interesting study. As a result, some of my comments and questions may stem from this. Below are my observations and suggestions to enhance the clarity and accuracy of the work:

  1. Title: It is recommended to include the type of study for greater clarity.
  2. Abstract:
    • Define abbreviations, such as EPT, upon first mention.
    • Headings should not be included in the abstract.
  3. Introduction: The stated objective mentions an association between variables, but no statistical analyses are presented to support this. It is suggested to either rephrase the objective or include the relevant analyses.
  4. Tables and Figures:
    • In Table 1, clarify the meaning of OR and CP for the readers.
    • Table 2 is currently placed in the methodology section but should be moved to the results section.
    • In general (unless table1), any table or figure placed outside the results section should be relocated accordingly.
  5. Justification for the selection of oligosaccharides: It is recommended to explain why these particular varieties were chosen over others.
  6. Precision in the use of statistical terminology: The text repeatedly states that an association has been found, but what has actually been identified are correlations. It is important to distinguish between these concepts, as they are not equivalent. This is one of the major concerns regarding the interpretation of the study’s findings.

Author Response

Reviewer 2

Comments and Suggestions on the Manuscript:

I appreciate the opportunity to review this article. I find it to be an interesting study. As a result, some of my comments and questions may stem from this. Below are my observations and suggestions to enhance the clarity and accuracy of the work:

  1. Title: It is recommended to include the type of study for greater clarity.
    • Added “an explorative trial” in the title
  2. Abstract:
    • Define abbreviations, such as EPT, upon first mention.
      • Yes, thanks for observing this mistake
    • Headings should not be included in the abstract.
      • Corrected
  3. Introduction: The stated objective mentions an association between variables, but no statistical analyses are presented to support this. It is suggested to either rephrase the objective or include the relevant analyses.
    • Thank you for your comment. It is somewhat unclear to us what the reviewer is asking for Following the praxis of Nutrients we only presented the statistical methods and results in the Methods and Results section.

  1. Tables and Figures:

    • In Table 1, clarify the meaning of OR and CP for the readers.
      • Yes, thanks for noticing this. OR is not an abbreviation, but could be misinterpreted as being one, as you pointed out. We have changed it and don’t use capital letters for “or”.
    • Table 2 is currently placed in the methodology section but should be moved to the results section.
      • Corrected
    • In general (unless table1), any table or figure placed outside the results section should be relocated accordingly.
      • Corrected
  1. Justification for the selection of oligosaccharides: It is recommended to explain why these particular varieties were chosen over others.
  • These 15 have been found to be dominating in previous HMO research performed by co-author Landberg and others, and thus may represent HMOs with the most potential clinical impact. This has been clarified in the text in the revised version..
  1. Precision in the use of statistical terminology: The text repeatedly states that an association has been found, but what has actually been identified are correlations. It is important to distinguish between these concepts, as they are not equivalent. This is one of the major concerns regarding the interpretation of the study’s findings.
    • Thank you for observing this. We have now corrected and use the term correlations for linear associations and otherwise association.
